# Machine learning predicts cancer subtypes and progression from blood immune signatures

**David A. Simon Davis[1], Sahngeun Mun[1], Julianne M. Smith[1], Dillon Hammill[2], Jessica Garrett[2], Katharine Gosling[2], Jason Price[2], Hany Elsaleh[3], Farhan M. Syed[1,4], Ines I. Atmosukarto[1,2], Benjamin J. C. Quah[1,4]***

**1** Irradiation Immunity Interaction Lab, Canberra, ACT, Australia, **2** Division of Genome Sciences & Cancer, John Curtin School of Medical Research, Australian National University, Canberra, ACT, Australia, **3** Radiation Oncology Department, The Alfred, Melbourne, VIC, Australia, **4** Radiation Oncology Department, Canberra Hospital, Canberra Health Services, Canberra, ACT, Australia

* ben.quah@anu.edu.au

**Data Availability Statement:** All relevant data are within the manuscript, its Supporting Information files and/or held in the Australian National University (ANU) DATA COMMONS repository at

## Abstract

Clinical adoption of immune checkpoint inhibitors in cancer management has highlighted the interconnection between carcinogenesis and the immune system. Immune cells are integral to the tumour microenvironment and can influence the outcome of therapies. Better understanding of an individual's immune landscape may play an important role in treatment personalisation. Peripheral blood is a readily accessible source of information to study an individual's immune landscape compared to more complex and invasive tumour bioipsies, and may hold immense diagnostic and prognostic potential. Identifying the critical components of these immune signatures in peripheral blood presents an attractive alternative to tumour biopsy-based immune phenotyping strategies. We used two syngeneic solid tumour models, a 4T1 breast cancer model and a CT26 colorectal cancer model, in a longitudinal study of the peripheral blood immune landscape. Our strategy combined two highly accessible approaches, blood leukocyte immune phenotyping and plasma soluble immune factor characterisation, to identify distinguishing immune signatures of the CT26 and 4T1 tumour models using machine learning. Myeloid cells, specifically neutrophils and PD-L1-expressing myeloid cells, were found to correlate with tumour size in both the models. Elevated levels of G-CSF, IL-6 and CXCL13, and B cell counts were associated with 4T1 growth, whereas CCL17, CXCL10, total myeloid cells, CCL2, IL-10, CXCL1, and Ly6C$^{intermediate}$ monocytes were associated with CT26 tumour development. Peripheral blood appears to be an accessible means to interrogate tumour-dependent changes to the host immune landscape, and to identify blood immune phenotypes for future treatment stratification.

## Introduction

Carcinogenesis is a complex and multi-layered process involving various cellular and tissue networks. Although tumours can be recognised by the immune system, resulting in their

https://dx.doi.org/10.25911/6153a8ab5747c
(which has the raw Flow Cytometry Standard (FCS) files).

**Funding:** This work was partially supported by the Radiation Oncology Private Practice Trust Fund, Canberra Health Services, Canberra, Australia. The funder provided support in the form of salaries and/or research materials for authors B.J.C.Q, D.A. S.D., S.M., J.S., F.M.S., I.I.A. but did not have any additional role in the study design, data collection and analysis, decision to publish, or preparation of the manuscript.

**Competing interests:** We have read the journal's policy and the authors of this manuscript have the following competing interests: I.I.A., J.P., and K.G. declare that they are employees of the biotechnology company Lipotek Pty Ltd. This does not alter our adherence to PLOS ONE policies on sharing data and materials. The remaining authors have declared that no competing interests exist.

growth suppression or elimination, they can also evolve to escape and/or suppress immune responses resulting in tumour growth and metastasis [1]. This interplay between tumour growth and the immune system is accompanied by specific perturbations to the immune landscape, manifested as changes to leukocyte frequencies and to the concentrations of immune soluble factors, both locally at the tumour site and systemically [2, 3]. Understanding the relationship between immune landscape changes and cancer subtype, disease progression and response to treatment has the potential to advance the development of new treatments, personalise therapies and improve outcomes.

Conventional cancer treatment strategies are comprised of the local modalities of surgery and radiotherapy, and the systemic approaches of endocrine treatment, cytotoxic chemotherapy, molecular targeted therapy and immunotherapy [4]. Whilst the latter two systemic therapies have, in recent years, brought the promise of dramatically improving cancer outcomes, their effect remains limited to only a subset of cancer patients [5]. Generic approaches to cancer treatment based only on tumour histology do not take into consideration the complexity of the cellular and tissue networks involved, and as such often result in variable outcomes. Moving beyond generic approaches and improving the outcome of novel immunotherapies requires robust preclinical interrogations to navigate the increasingly crowded arsenal of treatments. Preclinical models enable tumour-immune system networks to be studied in a controlled manner to establish clinically translatable workflows. Studying the peripheral immune system to identify tumour subtype-specific immune dysregulation signatures that could be associated with tumour growth and treatment outcome is one such avenue of focus in this approach.

The immunome–the genes and proteins that constitute the immune system and an ever-increasing number of immune cell types–is defined through complex patterns of antigen expression representing quantifiable metrics of an individual's immune landscape. Over a hundred different phenotypes of immune cells, representing the *cellular immunome*, have been identified. These include parental cell types such as T cells, B cells, natural killer (NK) cells, NK T cells, dendritic cells (DCs), granulocytes and monocytes. Each of these parental cell types can give rise to multiple subtypes defined by activation and/or suppressive functions. The *soluble immunome* consists of the chemokine system, which comprises nearly 50 chemokine ligands, and the cytokine system that includes over 30 glycoproteins regulating the functions of the immune system [6, 7]. Analysis of the complex interactions of these parameters is now possible, with machine learning (ML) techniques offering a means to generate models of multivariate data and facilitate the prediction of disease progression and treatment outcomes. ML techniques can also help with inference of underlying biological mechanisms based on explanatory algorithms [8].

This study examines a multiparameter immune phenotyping approach, using blood cellular and soluble immune signatures, in a tightly controlled preclinical environment of two well-established syngeneic tumour models: a triple-negative breast cancer, 4T1, and a colorectal cancer, CT26, both implanted subcutaneously in BALB/c mice. These tumour models were selected to allow the assessment of different immune signatures, as the 4T1 line is known to generate highly aberrant immune signatures [9, 10], whereas the CT26 line generates more subtle immune phenotype changes [11, 12]. To monitor changes to the systemic cellular and soluble immune signatures of tumour-bearing animals, a small volume of blood was obtained from the animals' tail veins in a minimally invasive, feature-rich and high-throughput strategy for clinical translation. Multiparameter flow cytometry was used to generate cell-surface immune signatures, while soluble immune profiles were readily obtained from the plasma using a bead-based immunoassay based on the same basic principles as sandwich immunoassays. This approach allows relatively high throughput generation of data and was coupled with

statistical modelling to make predictions and inferences about tumour outcomes and biology. Predictive modelling and feature ranking was performed using Random Forest models, in conjunction with SHapley Additive Explanations and correlation matrices, to make inferences about the underlying immune biology of the tumour models. This relatively simple strategy successfully generated reasonably accurate models that are able to (i) confirm the presence of a tumour, (ii) differentiate between tumour types and (iii) predict current and future tumour burden, and highlighted that both tumour models generate unique blood immune signatures.

This study aims to assess the utility of blood cellular and soluble immune signatures coupled with ML to predict cancer subtype and tumour progression in a tightly controlled preclinical environment. It provides evidence of potential clinical application of immune signature-based systemic immune phenotyping to improve overall cancer diagnosis and surveillance. The study also identifies key immune features for predictive modelling and possible candidate parameters for therapeutic intervention based on those models.

## Methods

To monitor changes to the systemic cellular and soluble immune signatures of tumour-bearing animals, a small volume of blood was obtained from the animals' tail veins in a minimally invasive, feature-rich and high-throughput strategy for clinical translation. Multiparameter flow cytometry was used to generate cell-surface immune signatures, while soluble immune profiles were obtained from the plasma using a bead-based immunoassay established on the same basic principles as sandwich immunoassays. This approach allows relatively high throughput generation of data and was coupled with statistical modelling to make predictions and inferences about tumour outcomes and biology. Predictive modelling and feature ranking was performed using Random Forest models, in conjunction with SHapley Additive Explanations and correlation matrices, to make inferences about the underlying immune biology of the tumour models. This relatively simple strategy successfully generated reasonably accurate models that are able to (i) confirm the presence of a tumour, (ii) differentiate between tumour subtypes and (iii) predict current and future tumour burden, and highlighted that both tumour models generate unique blood immune signatures.

### Animals

Female BALB/c mice aged between 6–10 weeks sourced from the Australian Phenomics Facility (ANU) were used throughout the study. Animals were fed *ad libitum*, housed in a specific-pathogen free environment and used under strict adherence to protocols approved by the institutional Animal Experimentation Ethics Committee (AEEC), ANU, under protocols A2017/43 and A2020/39. At experimental end points, animals were euthanised by cervical dislocation according to AEEC approved procedures.

### Cell lines

The 4T1-Luc2 (4T1) mammary carcinoma and CT26 colorectal carcinoma (kindly gifted by Dr. Aude Fahrer, ANU) cell lines were originally sourced from American Type Culture Collection (ATCC) and confirmed clear of pathogens by Cerberus Sciences (ISO 9001 Licence No. AU843-QC) before use in animals. Both adherent cell lines were cultured in RPMI-1640 (11875093, ThermoFisher Scientific) supplemented with 10% (*v/v*) fetal bovine serum (F8192, Sigma), 2mM L-glutamine (250300810, ThermoFisher Scientific), 10mM HEPES (15630080, ThermoFisher Scientific), 1mM sodium pyruvate (11360070, ThermoFisher Scientific) and 55 nM 2-mercaptoethanol (21985023, ThermoFisher Scientific), detached using warmed trypsin/

0.05% (*v/v*) EDTA solution (15400054, ThermoFisher Scientific) then passaged and maintained at up to 70–80% confluency.

## Tumour establishment

Tumour cells were injected at $1 \times 10^5$ cells in 50 μL of sterile normal saline solution subcutaneously in the right-hind flank of mice randomised across several housing cages. Fur around the injection site was removed by clippers prior to tumour inoculation. Tumours were left to grow for up to 14 days, and monitored daily to ensure wellbeing was maintained. In 21 of 98 of the 4T1-bearing mice, a single dose of the Src-inhibitor eCF506 (1914078-41-3, Sun-shine Chemical) at 0.1 (eC100), 1 (eC1000), or 10 (eC10000) mg/kg was administered i.p. 7 days post-tumour establishment, which appeared to have little, if any impact on the parameters assessed in the study, and so these mice were included to increase sample size (S1 File). At end-point, the mice were humanely sacrificed by cervical dislocation, and their tumours excised and weighed.

## Blood collection

At 7/8 (referred to as day 7) and 14 days post tumour establishment, mice were briefly heated (~4 minutes) under a lamp to promote vasodilation, placed in a restraint, their tail vein punctured with a 29G needle, and a 20 μL sample of blood collected into 4 μL of citrate-dextrose solution (ACD, Sigma) anticoagulant. A 5 μL sample of this blood was immediately used for antibody labelling and flow cytometry. The remaining blood was centrifuged at 16,000 x *g* for 10 minutes and 7 μL of plasma collected and stored in a sealed 96 well polypropylene microplate (249943, ThermoFisher Scientific) at -20˚C for future cytokine and chemokines measurements using the LEGENDplex assays.

## Immunophenotyping of blood leukocytes by flow cytometry

The 5 μL blood samples for cellular analysis were initially incubated for 10 minutes on ice in wells of a v-bottom 96-well microplate with 25 μL of 5 mg/mL TruStain FcX$^{TM}$ (anti-mouse CD16/32) antibody (101320, Biolegend) diluted in 1x RBC BD Pharm Lyse lysing buffer (555899, BD Bioscience). Samples were then incubated with 25 μL of 1x RBC BD Pharm Lyse containing fluorescent antibodies listed in S1 Table for 30 minutes on ice in the dark. In addition, 5000 Flow-Count Fluorospheres (7547053, Beckman Coulter) were spiked in to each sample with the fluorescent antibodies to allow enumeration of total cells per sample. Cells were then washed twice by resuspension in a total of 200 μL of PBS containing 5 mM EDTA, sedimentation by centrifugation at 300 x *g* for 5 minutes and flicking off supernatant. Samples were then resuspended in 50 μL of PBS containing 5 mM EDTA, 1% BSA (*w/v*) and 1 μg/ml of the dead cell dye Hoechst 33258 ready for flow cytometry.

## LEGENDplex assay

Frozen plasma samples were thawed on ice, then assayed using the Macrophage/microglial (Mac/Mic) 13-plex LEGENDplex kit and the Proinflammatory (Proinflam) 13-plex LEGENDplex Kit (740451 and 780846, Biolegend). Assay methods were as described by the manufacturer, except the assay was scaled down to use 6 μL of sample/standards for each kit as follows. Seven μL of each plasma sample was diluted in 7 μL of kit assay buffer and 6 μL of this mix (or 6 μL of pre-titrated kit standard) was added to 12 μL of kit capture beads (pre-diluted 1:1 (*v/v*) with assay buffer) in a v-bottom 96-well microplate, and incubated with shaking for 2 hours. Beads were then pelleted at 250 x *g* for 5 minutes and the supernatant flicked off. Beads were

washed with 50 μL of kit wash buffer and pelleted and supernatant removed as above. Twelve μL of kit biotinylated detection antibodies (pre-diluted 1:1 (*v/v*) in assay buffer) were then added to beads, then beads resuspended by pipetting and the mixture incubated with shaking for 1 hour at room temperature. Six μL of kit streptavidin-PE was then added to the mixture, which was incubated with shaking for a further 30 minutes. Beads were then pelleted and washed as described above and resuspended in 40 μL of kit 1x wash buffer ready for flow cytometry.

## Flow cytometry

Flow cytometry was performed on a BD LSRII (BD Bioscience) flow cytometer with FACS-Diva, with quality assurance performed before each experimental run using BD FACSDiva Cytometer Setup and Tracking (CS&T) beads (655051, BD Bioscience). Application Settings were applied to standardise fluorescence intensity readings between experiments and these were monitored using Sphero$^{TM}$ 8-peaks Rainbow Beads Fluorescence (110620, BD Bioscience). Voltages were initially setup using unlabelled RBC-lysed blood leukocytes for cellular analysis and LEGENDplex Raw Setup beads (as described by the manufacturer). BD Comp-Beads (552843, BD Bioscience) were labelled with selected antibodies (S1 Table) as described by the manufacturer and used as compensation controls for cellular analysis. Cell samples were acquired until a total of 2000 Flow-Count Fluorosphere beads were collected based on side scatter (log) and forward scatter (linear) plot gating. LEGENDplex beads were acquired to a total of 4000 beads. Raw Flow Cytometry Standard (FCS) files of the data are available upon request at the ANU DATA COMMONS repository (https://dx.doi.org/10.25911/6153a8ab5747c).

## Flow cytometry analysis

Blood cells and LEGENDplex beads were analysed using FlowJo v10 software (BD Bioscience). A combination of manual gating and unsupervised Fast Interpolation-based t-distributed Stochastic Neighbour Embedding (FIt-SNE) analysis was use to delineate leukocyte populations, which were then named based on this analysis (S1 Fig). LEGENDplex beads were gated for each analyte as describe in S2 Fig and median PE fluorescence-intensity generated for each bead analyte. Data was then normalised as describe below for analysis.

## Data normalisation and processing

To reduce the influence of inter-experimental variability on conclusions, data was normalised at several levels. Firstly, cell numbers in each flow cytometry acquisition were normalised to counting beads spiked into the sample, with each sample normalised to 2000 counting beads (a fifth of the spike load), to give the number of cells in ~2 μL of blood ("counting bead normalised" values). Secondly, these normalised counts were normalised to the mean counts from the blood of non-tumour bearing control animals within each experiment. These "nil normalised" values were used in machine learning pipelines. To get "normalised cell counts" per 2 μL of blood, for an estimate of the overall cells across the groups, the "nil normalised values" were multiplied to the overall mean of the "bead normalised cell count" from all non-tumour-bearing animals for each feature.

For the LEGENDplex assays, the raw PE median analyte values were normalised as a ratio to the mean PE median analyte values from the blood of non-tumour bearing control animals within each experiment. These "nil normalised" values were used in machine learning pipelines. To get "normalised plasma concentrations", the "nil normalised" values were multiplied to the overall mean PE median from the blood of non-tumour bearing control animals, and

concentrations interpolated using mean standard curves pooled from all experiments, with Hyperbola, 5-parameter logistic regression (5PL) and Random Forest models employed. Since 5PL models failed for many data points and Random Forest resulted in non-gaussian multi-cluster distributions, hyperbola models were used as they overcame these issues. t-distributed stochastic neighbour embedding (t-SNE) unsupervised clustering was used to monitor experimental clusters within the pooled data and helped to confirm experimental cluster minimisation using the normalisation approach. All raw and calculated data are in S1 File.

## Supervised machine learning

Supervised machine learning was performed using Orange 3 software. Random Forest modelling used 500 trees, with a maximum tree depth of 3, a maximum number of features considered at each node was 4 (except when considering smaller feature numbers in which case the hyperparameter changed accordingly), subsets smaller than 5 not split, and balanced class distribution enabled in the case of classification learning since data groups were unbalanced. Missing data (that included a single sample without the 13 Proinflammatory LEGENDplex panel) was imputed using the "hot deck" 1-NN learner, which replaces the missing values with the values from the most similar example (as implemented in Orange 3 software). Initially, a learning curve was generated by plotting progressively smaller data set size (randomly generated from the entire data set) against modelling skill (assessing classification of the tumour subtype; 4T1, CT26 and Nil) to evaluate if the data set size was sufficient for the outcomes targeted (S3 Fig). This revealed the data set size at 20% appeared to plateau in modelling skill, suggesting data size was sufficient for the outcomes targeted. For the rest of the study, Random Forest model training was performed and cross-validated on 100%, 80% and/or 60% of randomized sample data and tested on any remaining data. The training data was validated using leave-one-out cross-validation. Feature ranking was done using Random Forest (built in to the Random Forest model in Orange 3 software) and the explain model function on Orange 3, which uses the SHapely Additive exPlanation (SHAP) to explain feature importance. Feature number and model fitting was optimised for classification predictions based on area under curve of the receiver operating characteristics (AUC; to assess separability of the classes), classification accuracy (CA; proportion of correct classification), precision (ratio of correct positive prediction to all predicted positive), recall (ratio of correct positive prediction to actual positive), and F1 score (weighted average of precision and recall) classification scores and for regression using, Mean Squared Error (MSE), Mean Absolute Error (MAE) and Root Mean Squared Error (RMSE) scores. Each train and test modelling was done a minimum of 3 times to assess variability. Once optimal features were assigned based on the above, the final predictions were modelled on all the data and results displayed using cross-validation via leave-one-out on the entire data set, either as a confusion matrix for classification analysis, or a bivariate plot to actual values for regression analysis (with Pearson correlation coefficient reported and associated p values calculated using prism software). Orange 3 workflows are provided in S2 File (for classification workflow) and S3 File (for regression workflow).

## Statistical analysis

For means comparisons between Nil, CT26 and 4T1 cohorts, data was transformed using the formula $Y = Log(Y+0.0001)$ to help normalise distributions and equalise variance, and then assessed by 2-way ANOVA using GraphPad Prism software. Multiple comparisons were performed between the cohorts for each feature using Tukey correction and p values reported to test the null hypothesis that the means are equal. For analysis of important features in tumour size, a bivariate correlation matrix was designed using the top assigned features from the

machine leaning pipeline described above and Spearman's correlation coefficients and associ-ated p values determined using R. To determine interaction of top assigned features, a distance matrix was constructed using the absolute Spearman's coefficients and global absolute Spear-man distances summarised using multidimensional scaling with network lines (at maximum levels) using Orange 3 software (see S4 Files for Orange 3 workflow).

## Results

### Composition of blood immune features in cancer models reveals unique tumour immune phenotypes

To characterise tumour-bearing animal blood immune profiles, a 4T1 breast cancer cell line tumour or a CT26 colorectal cancer cell line tumour was established subcutaneously in the right-hind flank of BALB/c mice. Animals with no tumours were used as controls (Nil), bench-marking the 'normal' immune landscape. Tumours were left to establish and grow for 14 days and immune features assessed from a single drop of blood taken at 7 (D7) and 14 (D14) days post tumour establishment (Fig 1A). A total of 180 animals were included in the study with animal cohorts described in Fig 1B. Absolute leukocyte count per unit volume of blood were assessed using flow cytometry (S1 Fig). The cell populations were delineated using 17 leuko-cyte-reactive mAbs and identified using manual gating cross-checked with unsupervised dimensional reduction. The strategy also included simple light-scattering profiles to delineate lymphocytes and myeloid cells, to determine if this simple approach would be beneficial for the study aims. Blood plasma cytokine and chemokine concentrations were also assessed using two 13-plex LEGENDplex kits (S2 Fig). At the end-point (D14), solid tumours were extracted and weighed and revealed highly variable tumour mass in the two tumour models, ranging from 10 mg to >800 mg (Fig 1C).

To gain an overall impression of the blood immune landscape, the means of blood leuko-cytes and plasma factor composition were quantified across the 3 groups from the 180 animals at both D7 and D14 time points (Fig 1D), and differences further highlighted by normalising the underlying data to the mean values from Nil animals to give fold-change above normal lev-els (S4 Fig). This revealed a large increase in leukocytes in the blood of 4T1-bearing mice, com-pared to the Nil mice, a difference that increased further over time (Fig 1D). This was largely driven by expansion of myeloid cells but also a subtler trend of lymphocyte increase. In con-trast, there was only a slight trend of myeloid cell increase and a concomitant trend of lympho-cyte decrease in CT26-bearing animals, which became more exaggerated over time. The changes in myeloid cells in both models was largely attributed to an expansion of neutrophils and monocytes. Expansion of other minor myeloid cell populations was also apparent (Fig 1D). The initial increase in lymphocytes in 4T1-bearing mice at D7 was mostly due to an increase in B cell count, which reversed with a decrease from normal at D14 and was compen-sated for by slight increases in CD4 T cells, CD8 T cells and NK cells at this later time point. The decrease in blood lymphocytes in CT26-bearing mice was mainly attributed to diminish-ing circulating B cells. There were also changes in minor subpopulations of leukocytes in tumour-bearing animals not obvious in the compositional data due to their small numbers; these included changes to CD4 T regulatory cells, DC, macrophages and PD-L1-expressing myeloid cell populations (S4 Fig). In terms of plasma factor composition, there was a notable increase in macrophage/microglial factors in 4T1-bearing mice at D7, mainly ascribed to a large increase in G-CSF, which decreased at D14, although was still several-fold above normal levels (Fig 1D). Mice with 4T1 tumours also had a subtler increase in CXCL13 relative to nor-mal levels, and a subtle increase in IL-6 and a subtle decrease in CXCL1 compared to CT26-bearing animals (Fig 1D and S4 Fig). In CT26-bearing mice, there was a slight rise in the

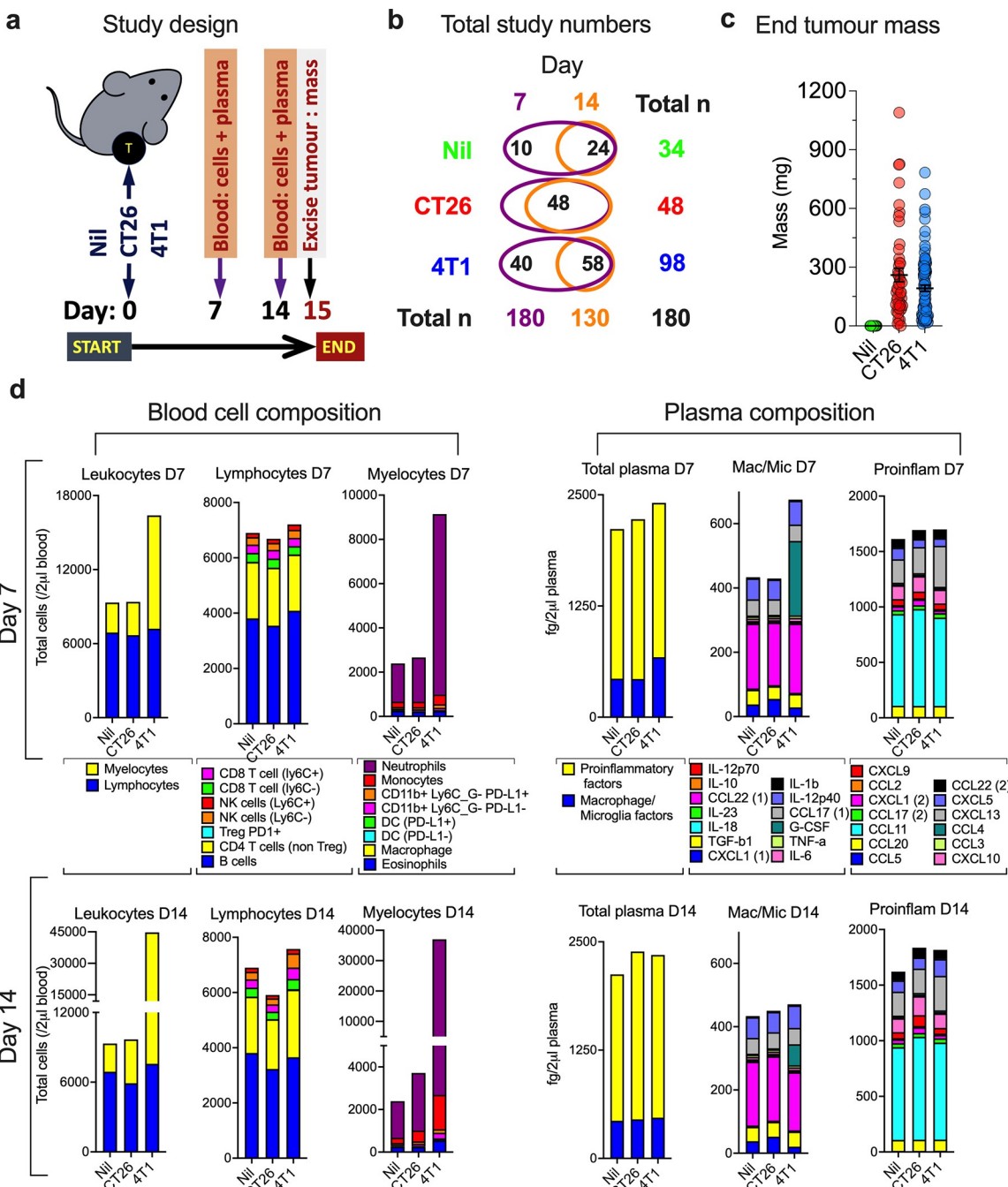

**Fig 1. Blood immune phenotyping in animal tumour models.** CT26 or 4T1 tumours were established and grown in female, BALB/c mice for 14 days, with blood immune phenotype determined by flow cytometry 7-(D7) and 14-(D14) days post tumour implantation, and tumours excised and weighed at end-point (D14). Animals with no tumours (Nil) were used as control to provide normal blood immune phenotype (**a**). A total of 180 animals were included in the study, and animals were randomly allocated to groups at D0, as indicated in (**b**). End-point (D14) CT26 and 4T1 tumour mass are shown in (**c**) with mean and SEM overlayed in black. A 20 μl of blood sample from each animal was phenotyped for immune cell populations (using cell surface marker labelling and reported as normalised cells per ~2μl of blood) and plasma analytes (using two LEGENDplex screening kits and reported as approximations of blood concentrations) at D7 and D14 by flow cytometry. Blood cell compositions at D7 (*top*, *left panel*) and D14 (*bottom*, *left panel*), and plasma analyte compositions at D7 (*top*, *right panel*) and D14 (*bottom*, *right panel*) in Nil, CT26- and 4T1-tumour bearing animals, respectively, are shown in (**d**). Cell data in (d) was reported as total absolute mean cell count of each population being a subset of upstream lineages. Plasma analytes were reported as a subset of total mean of analytes in the two LEGENDplex screening kits, which included the macrophage/microglial 13-plex kit (Mac/Mic) and the proinflammatory 13-plex kit (Proinflam). Three analytes overlapped in the kits, namely CCL22, CXCL1 and CCL17, and are labelled with a (1) if from the Mac/Mic panel or (2) if they are from the Proinflam panel.

proinflammatory factors in plasma, which increased marginally over time, and appeared to be due to subtle changes in a number of factors such as CCL11, CXCL1, CXCL9 and CXCL10 (Fig 1D). These changes, however, were not statistically significant from control animals (S4 Fig).

## Classification of cancer models using blood immune signatures

From these initial results, it was clear that 4T1 and CT26 tumour growth results in aberrant blood immune parameters in mice, with some common changes (such as neutrophil and monocyte expansion), but also tumour-specific changes (such as the plasma factor changes), while overall changes appear to be subtler in CT26-bearing mice. To investigate how these changes might be used to predict tumour outcomes, supervised ML was used on the normalised data (S4 Fig). Random Forest was chosen as our learner, since it could be used for prediction of both our classification (tumour subtype) and regression (mass of current and future tumours) questions and has in-built feature ranking of importance in predictions allowing feature reduction and biological inference [13].

After hyperparameter tuning, Random Forest was initially used to investigate if blood immune phenotypes were unique enough to classify whether animals had no tumour (Nil), or had a CT26 or a 4T1 tumour. Our approach was to train and test the model using progressively reduced numbers of blood immune features, sorted based on importance rank. We scored the model using several prediction classification indicators (S5 Fig and Fig 2A). To train and test the model, we used data from both D7 and D14 time points, to see if there were features that could be used across time to classify a tumour-subtype. From this we found the modelling was stable and had congruent scores in both the training and test data sets across a range of features fed into the model. However, the minimum feature number needed to maintain this was 5, suggesting 5 key features could result in accurate predictions (Fig 2A). Looking at the top 21 Random Forest ranked features, there were several that were highly ranked at both the D7 and D14 time points (Fig 2B). Overall, the 5 highest ranked immune features, in descending order, were G-CSF, neutrophils, total myeloid cells, monocytes and total leukocytes. To look at how these features contributed to the model in more detail, we used the **SH**apley **A**dditive ex**P**lanations (SHAP) algorithm [8] (S5 Fig). SHAP highlighted the contribution of these 5 features: generally, as they increased, they tended to suggest a 4T1 phenotype, while there was a more complex relationship in distinguishing Nil from CT26-bearing animals. We performed dimensional reduction using t-distributed Stochastic Neighbour Embedding (t-SNE) to see if these 5 features could cluster tumour classes better than all features combined (Fig 2C). Using this unsupervised approach showed the 5 top-ranked features appeared to separate the tumour classes better than all features combined, particularly the 4T1 class. Therefore, we generated the final model incorporating these features from both time points (Fig 2D). This resulted in successful classification of all 4T1-bearing animals and most of the CT26-bearing (CA ~80%) and Nil (CA ~75%) classes (2 of each being misclassed as the other out of 72 individuals in these classes). The 5 features showed the capacity to predict class at each time point alone, but generally predicted and separated classes best at the later time point (Fig 2E and 2F). Finally, looking at their quantity in the blood of all animals showed that, while these features were all significantly higher in 4T1-bearing animals compared to CT26-bearing and Nil animals, only neutrophils and monocytes showed a significant increase in CT26-bearing mice compared to Nil (while still being lower than in 4T1-bearing mice) (Fig 2G). This highlights the association of myeloid factors with tumour presence and their potential use in tumour classification and may also suggest an underlying association of G-CSF, neutrophils and monocytes in the development of some tumours.

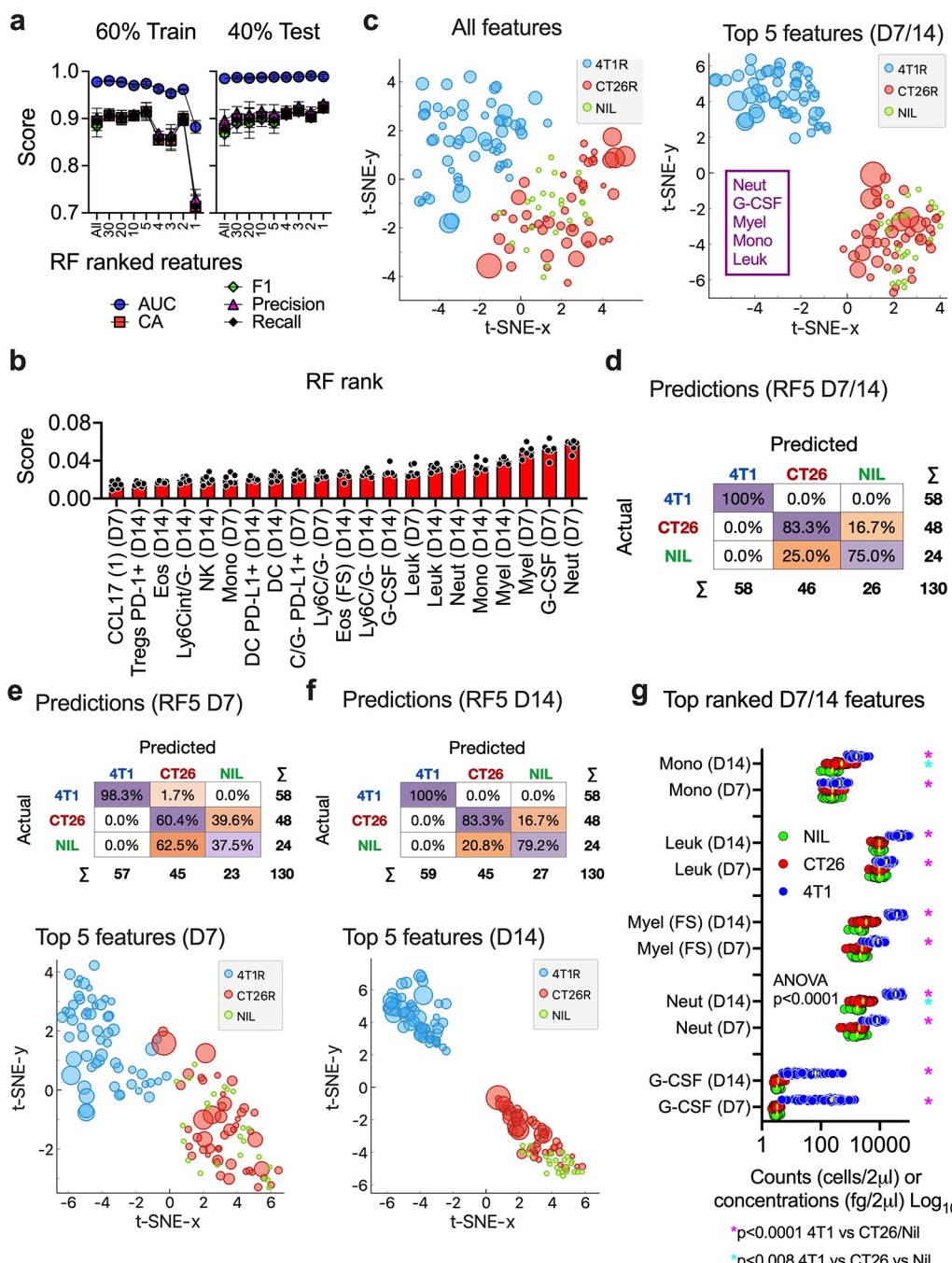

**Fig 2. Tumour classification using blood immune phenotype.** Normalised blood immune features (S4 Fig) taken from the 130 animals that had both D7 and D14 blood samples (Fig 1B), were used in Random Forest modelling to predict presence of tumour and tumour subtype (targets class being Nil, 4T1 and CT26). The model was trained initially on 80% (S5 Fig) and 60% of data, cross-validated using leave-one-out and tested using the remaining data. Modelling was done on a progressively smaller number of features, from lowest to highest ranked based on in-built Random Forest importance for class determination, and the process repeated 3 times. Model performance was assessed by several classification indicators, including area under curve of the receiver operating characteristics (AUC; to assess separability of the classes), classification accuracy (CA; proportion of correct classification), precision (ratio of correct positive prediction to all predicted positive), recall (ratio of correct positive prediction to actual positive), and F1 score (weighted average of precision and recall) with values being from 0 to 1 (and toward the latter being the best) (**a**). The Random Forest feature importance scores for classification of the top 21 features (ranked from lowest to highest) from the modelling are show in (**b**) from n = 6 modelling repeats. Based on peak modelling performance (S5 Fig), the top 5 features from both time points were compared with all features in t-distributed stochastic neighbour embedding (t-SNE)

unsupervised clustering to highlight capacity of reduced features to maximise class distinction based on the overlap of groups, with dot sizing representing relative end-point tumour mass to assess for how this relates to clusters (**c**). These top 5 features from both time points were used to generate the final classification modelling, which was performed on the entire data set and assessed using leave-one-out cross-validation and results shown as a confusion matrix of all animals (**d**). The top 5 features from D7 (**e**) or D14 (**f**) samples were also used in modelling (presented as confusion matrices) and t-SNE analysis to highlight time differences. The top 5 features were plotted as estimated quantities in blood for all animals at each time point (Fig 1B) and their means and SEM displayed in yellow, and means equality tested using 2-way ANOVA on Log (y+0.0001) transformed data and multiple comparison with Tukey's correction shown (**g**).

## Model fitting of CT26 tumour size using blood immune signatures resulted in moderate predictability

We next wanted to see if underlying blood immune signatures could be used to predict tumour size and growth, which are often fundamental to prognosis. To do this we used the D14 end-point tumour mass as the target outcome. We first assessed whether blood immune signatures could predict current and future CT26 tumour mass with D14 and D7 blood data respectively. As with the classification approach, we trained and tested the model using progressively reduced numbers of blood immune features sorted based on importance rank, but scored the model using several regression prediction indicators (S6 Fig and Fig 3). Testing if D14 blood could predict current tumour mass, we found Random Forest modelling was stable and had similar scores in both the training and test data sets across a range of features fed into the model; however, the minimum feature number to maintain this was 3, suggesting 3 key features could result in optimal current tumour mass predictions (S6 Fig and Fig 3A). Myeloid cell populations ranked high in modelling (Fig 3B), with Ly6C$^{intermediate}$ monocytes, total myeloid cells, and PD-L1-expressing Ly6C$^-$Ly6G$^-$ (PD-L1$^+$) myeloid cells contributing prominently to the model based on SHAP values (Fig 3C). Mice with higher numbers of these cells in the circulation typically had bigger tumours. We therefore generated the final Random Forest model with these 3 features to predict the current mass of CT26 tumour, which resulted in a significant moderate linear correlation with the actual mass (Fig 3D).

Testing if D7 blood immune features could predict future D14 CT26 tumour mass, we found the minimum feature number to maintain modelling peaked at 10 features (S6 Fig and Fig 3E). While myeloid cells were an important feature, there were also several plasma immune factors, notably CCL17, CXCL10, CXCL1 and CXCL13, that had high importance (Fig 3F and 3G). However, from the SHAP explanations, it was apparent that extreme values of many of these features in only a few animals impacted on the model, suggesting poor general association with tumour size (Fig 3G). Generating the final Random Forest model with these 10 features to predict the future mass of CT26 tumour resulted in a significant moderate linear correlation with the actual mass (Fig 3H).

## CT26 tumour mass prediction modelling suggests several key blood immune features associate with tumour development

SHAP values of immune features predicting CT26 tumour mass suggest several features have a relationship with tumour size that together allow moderately strong tumour mass predictions to be made. To investigate this in more detail, and possibly infer some immune mechanisms supporting tumour growth, a correlation matrix was plotted of the 5 key predictive features from both D7 and D14 blood samples, and their monotonic relationship reported via Spearman's correlation coefficient (Fig 3I). While there appeared to be significant weak-to-strong relationships among the 5 key D7 features, only CXCL10 had a significant, but weak, relationship with end-point CT26 tumour mass. In contrast, 4 of 5 key D14 features of tumour growth

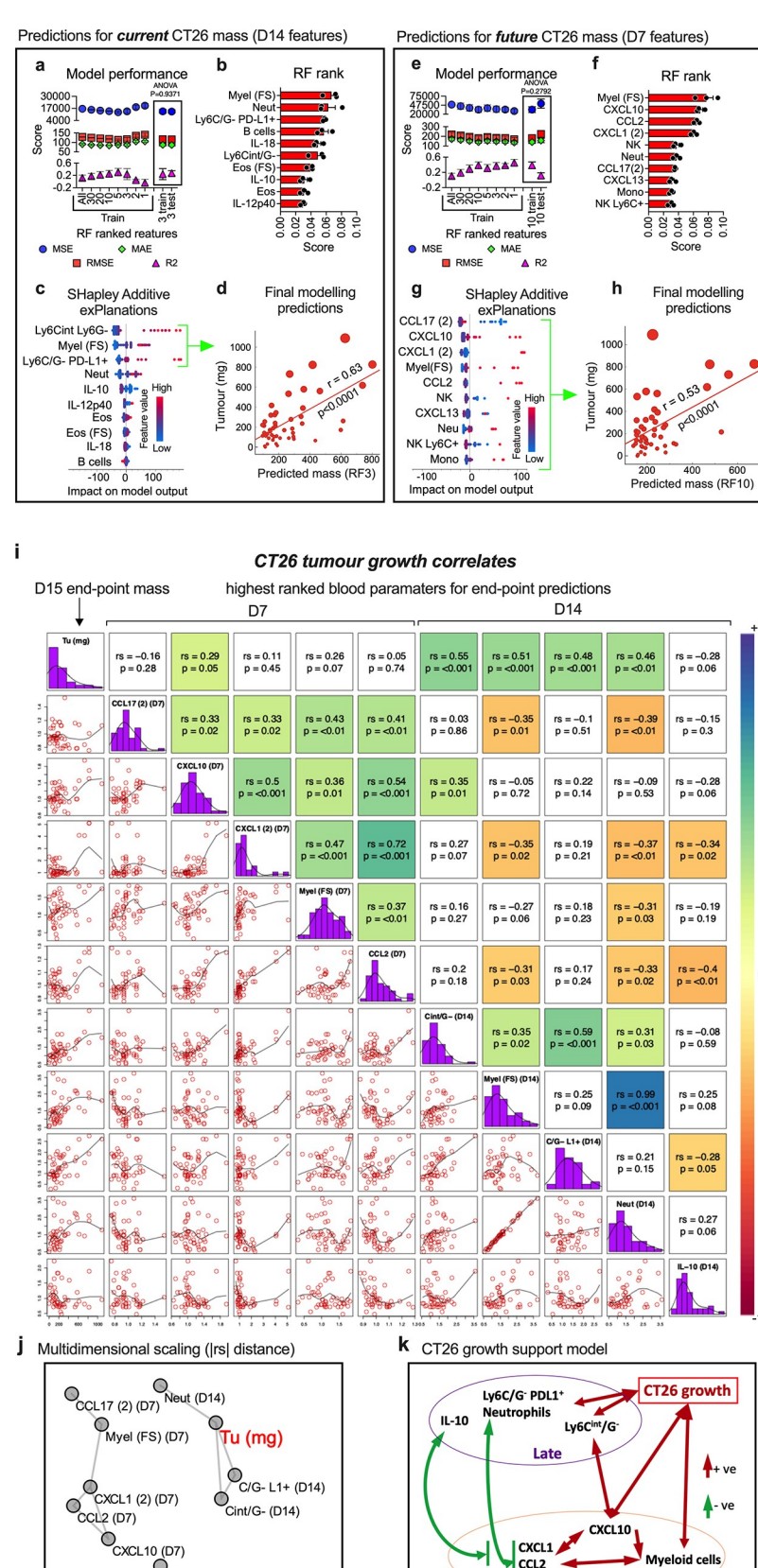

**Fig 3. Predicting CT26 tumour size and growth using blood immune phenotypes.** Normalised blood immune features (S4 Fig) taken from the 48 CT26-bearing animals that had both D7 and D14 blood samples (Fig 1B), were used in Random Forest modelling to predict CT26 tumour size at D14. The model was trained initially on 100%, 80% and 60% of data (S5 Fig) and cross-validated using leave-one-out and tested using the remaining data. Modelling was done on a progressively smaller number of features, from lowest to highest ranked, based on in-built Random Forest importance, and the process repeated 3 times (mean and standard error of mean shown). Model performance was assessed by several regression indicators, including the error scores, Mean Squared Error (MSE), Mean Absolute Error (MAE) and Root Mean Squared Error (RMSE) (which we hoped to minimise), and the coefficient of determination score R2 (which we hoped to maximise). Initially, D14 tumour size was used as the target using D14 blood samples to assess if blood immune features could predict *current* tumour size (**a, b, c** and **d**). Then D14 tumour size was used as the target using D7 blood samples to assess if blood immune features could predict *future* tumour size (**e, f, g, h**). Model performance was summarised showing the 60%:40%, training:testing split and equality of test and train performance score means (using the top assigned features) assessed using ANOVA (**a**) and (**e**). The Random Forest feature importance scores for regression of the top-10 features from the modelling are show in (**b**) and (**f**), and the SHAP scores of these shown in (**c**) and (**g**). Based on peak modelling performance, the top-3 features from D14 blood data (**d**) or top-10 features from D14 blood data (**h**) were used to generate the final regression modelling to predict *current* and *future* tumour mass respectively. Final modelling was performed on the entire data set and assessed using leave-one-out cross-validation and predicted mass of tumour plotted against actual tumour mass (the *y*-axis), in scatter-plots with dot sizing representing actual end-point tumour mass to assess for how this relates to any clusters, and the linear relationship assessed using Pearson correlation coefficient (r) and associated two-tailed *p*-values (**d** and **h**). Using the top-5 ranked features at each time point a correlation matrix was constructed, which displayed all pair-wise bivariate plots with loess curve fitting (lower-left half), feature names and distributions (diagonal) and Spearman's correlation coefficient (rs) with associated *p*-values to test for monotonic relationships (upper-right half), which was also colour-scaled based on rs values that had *p*-values <0.05) (**i**). A distance matrix of the absolute rs (|rs|) from the correlation matrix was calculated and distances plotted in 2D using multidimensional scaling (**j**) and a model of the interactions summarised in (**k**).

had significant direct association with tumour mass and one another. The relationship of the key D14 features and the key D7 features was complex, with both negative and positive significant relationships (Fig 3I). Generally, CCL17 weakly and positively correlated with early factors of tumour grow (CXCL10), but then weakly and negatively correlated with late factors of tumour growth (myeloid cell populations); CXCL1 and CCL2 acted like CCL17 in this respect. To summarise all these interactions, the distance of absolute values of the Spearman's correlations was plotted using multidimensional scaling, which shows the global relationships of the features and tumour size in 2 dimensions (Fig 3J). This emphasised the key association of D14 neutrophils, PD-L1$^+$ myeloid cells and Ly6C$^{intermediate}$ monocytes with CT26 tumour size, and a more distant relationship with D7 myeloid cells, CCL17, CXCL1, CCL2, CXCL10 levels, and D14 IL-10 level. From this we could postulate that CCL17, CXCL1 and CCL2 act early and in a similar way to indirectly help tumour growth, possibly by upregulating CXCL10 production and myeloid cell expansion, which act more directly on tumour growth. These correlations change with time, with early low expression of CCL17, CXCL1 and CCL2 eventually promoting myeloid cell development that maintains/promotes larger tumours. A possible model of blood immune features associated with CT26 tumour growth is depicted in Fig 3K.

## Model fitting of 4T1 tumour size using blood immune signatures resulted in strong predictability

To determine if 4T1 tumour growth could also be predicted by blood immune phenotype, a similar work flow to the above was employed. First, we tested if D14 blood features could predict current 4T1 tumour mass. Random Forest modelling was stable and had similar scores in both the training and test data sets across a range of features fed in to the model, with scores peaking with 3–5 features (S7 Fig and Fig 4A). Myeloid cells and neutrophils ranked highest in modelling, and high values of these associated with larger tumours (Fig 4B and 4C). B cell count was also among the 3 top ranked features and, generally, lower B cell numbers correlated with a higher 4T1 tumour mass (Fig 4B and 4C). There was a more complex relationship

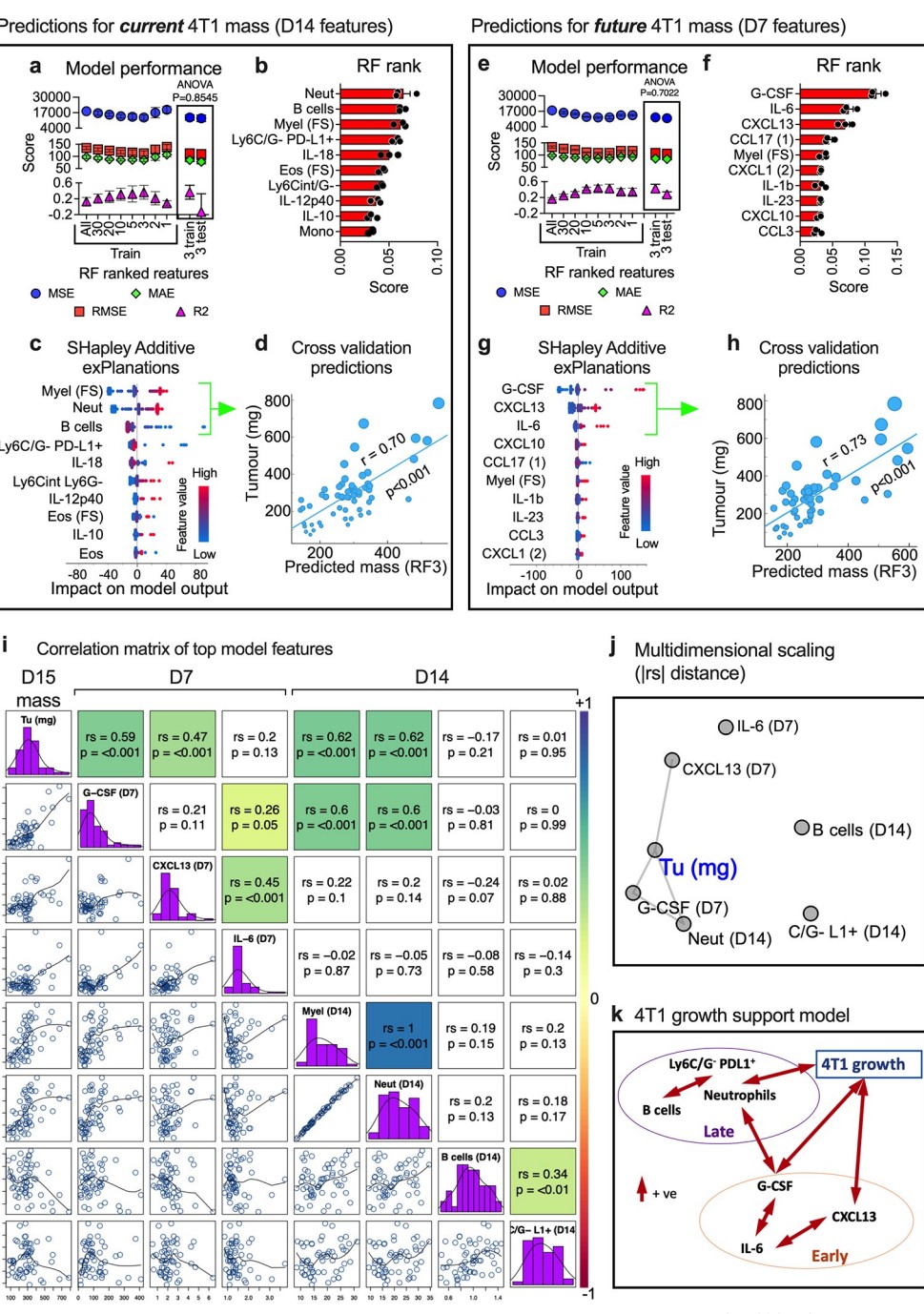

**Fig 4. Predicting 4T1 tumour size and growth using blood immune phenotypes.** Normalised blood immune features (S4 Fig) taken from 58 4T1-bearing animals that had both D7 and D14 blood samples (Fig 1B), were used in Random Forest modelling to predict D14 4T1 tumour size. The modelling and assessment was performed as described in Fig 3. Initially, D14 tumour size was used as the target using D14 blood samples to assess if blood immune features could predict *current* tumour size (**a, b, c** and **d**). Then D14 tumour size was used as the target using D7 blood samples to assess if blood immune features could predict *future* tumour size (**e, f, g, h**). Model performance was summarised showing the 60%:40% training:testing split and equality of test and train performance score means (using the top assigned features) assessed using ANOVA (**a** and **e**). The Random Forest feature importance scores for regression of the top-10 features from the modelling are shown in (**b**) and (**f**), and the SHAP scores of these shown in (**c**) and (**g**). Based on peak modelling performance, the top-3 features from D14 blood data (**d**) and D7 blood data (**h**) were used to generate the final regression modelling to predict *current* and *future* tumour mass respectively. Final modelling was performed on the entire data set and assessed using leave-one-out cross-validation and predicted mass of tumour plotted against actual tumour mass (the *y*-axis), in scatter-plots with dot sizing representing actual end-point tumour

mass to assess for how this relates to any clusters, and the linear relationship assessed using Pearson correlation coefficient (r) and associated two-tailed *p*-values (**d** and **h**). Using the top ranked features at each time point a correlation matrix was constructed, which displayed all pairwise bivariate plots with loess curve fitting (lower-left half), feature names and distributions (diagonal) and Spearman's correlation coefficient (rs) with associated *p*-values to test for monotonic relationships (upper-right half, which was also colour-scaled based on rs values that had *p*-values <0.05) (**i**). A distance matrix of the absolute rs (|rs|) from the correlation matrix was calculated and distances plotted in 2D using multidimensional scaling (**j**) and a model of the interactions summarised in **k**.

between the next highest ranked feature, PD-L1$^+$ myeloid cells and the model, with lower numbers of these cells associating with high and lower tumour size. Using the top 3 key features in the final model resulted in predictions with a significant strong linear relationship with actual current 4T1 tumour mass (Fig 4D).

Testing if D7 features could be used to predict future 4T1 tumour mass at D14, the Random Forest modelling had peak performance with ~3 key features (S7 Fig and Fig 4E). The 3 main model drivers were plasma G-CSF, CXCL13 and IL-6 levels, with higher plasma amounts of these factors generally associating with larger 4T1 tumours (Fig 4F and 4G). Using these 3 features in the final model resulted in predictions with a significant strong linear relationship with actual future 4T1 tumour mass (Fig 4H).

## 4T1 tumour mass prediction modelling suggests a few key blood immune features associate with tumour development

SHAP values of immune features predicting 4T1 mass suggest there were 6–7 features that have a relationship with tumour size that together have strong tumour mass prediction value. A correlation matrix was plotted of the 7 key features collectively from D7 and D14 blood samples, and their monotonic relationship reported via Spearman's correlation coefficient (Fig 4I). These interactions were also summarised using multidimensional scaling to plot the distance matrix of the Spearman's correlations' absolute values (Fig 4J). From this it appeared that plasma G-CSF level associated directly with 4T1 tumour mass and blood neutrophil count; the latter also associated directly with 4T1 tumour growth. Plasma CXCL13 level also had a direct positive association with 4T1 tumour growth, but did not appear to correlate with plasma G-CSF level or myeloid cell counts. In contrast, plasma IL-6 level had no direct association with 4T1 tumour size, but correlated positively with factors that did, namely plasma G-CSF and CXCL13 levels. The role of B cells and PD-L1$^+$ myeloid cells is unclear using monotonic measures, suggesting that if they do have a role, it is more complex. From this, we could postulate and form a model (Fig 4K) that IL-6 acts to promote CXCL13 and G-CSF production, which may act independently to aid 4T1 growth, and that G-CSF also promotes neutrophil expansion that supports 4T1 tumour growth.

## Summary of key tumour mass associated features

From the above analysis, there were a number of features that were important in modelling predictions for tumour growth and that associated directly or indirectly with specific tumour size. The estimated quantities of these features in blood and their comparisons between the models are summarised in Fig 5. From these pairwise comparisons it is apparent that most of the blood features that are important for modelling and correlating with CT26 growth, namely CCL17, CXCL10, total myeloid cells, CCL2, IL-10 and PD-L1$^+$ myeloid cells, were not significantly different from the healthy levels in Nil mice (Fig 5). Indeed, of the identified important features for CT26 growth, only CXCL1, Ly6C$^{intermediate}$ monocytes and neutrophils had quantities in CT26-bearing animals significantly different from normal blood of Nil animals, and in all cases higher than normal.

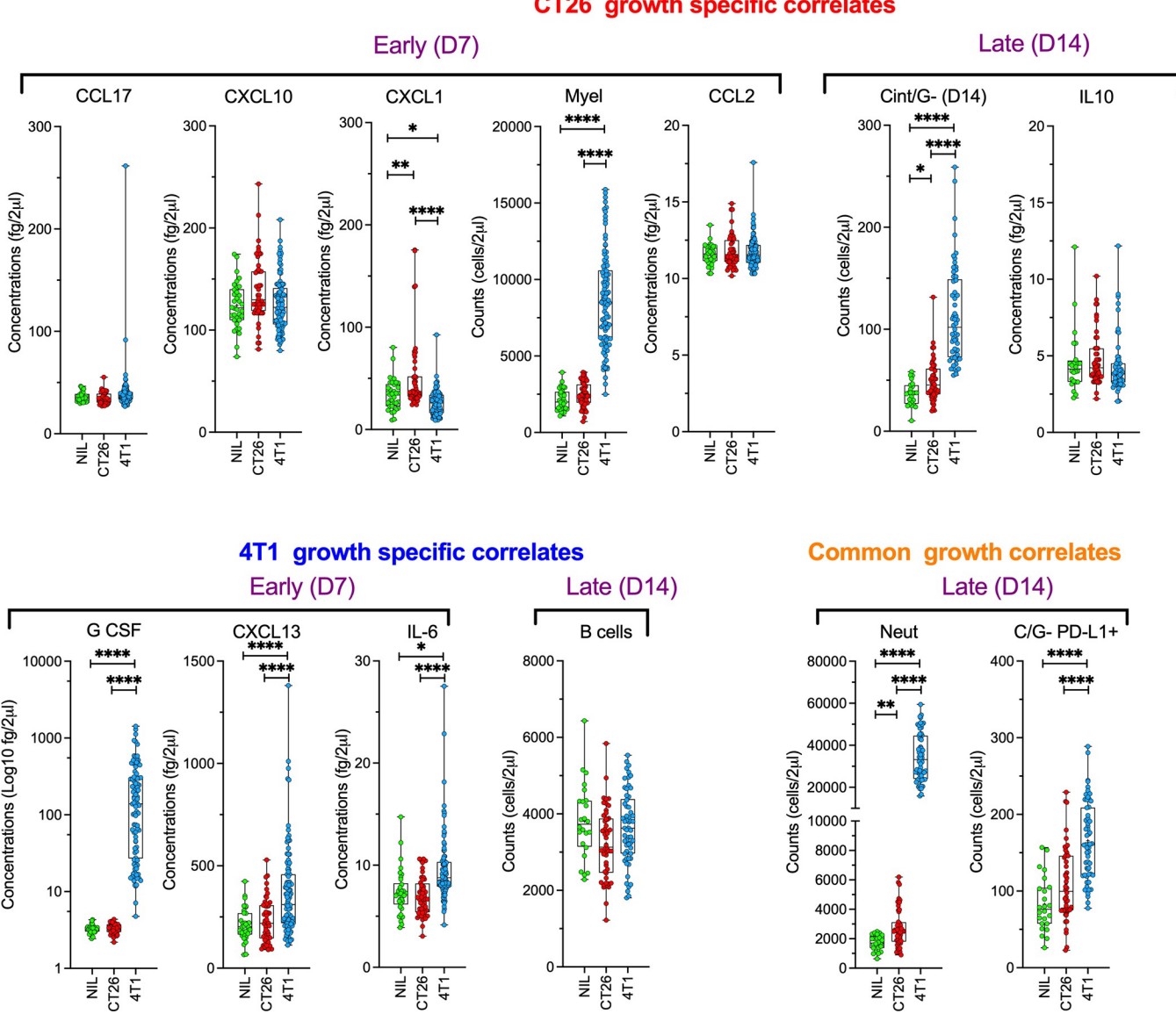

**Fig 5. Blood immune features associating with tumour growth.** The quantities in blood of key immune features associated with both CT26 and 4T1 tumour growth from both D7 (early) and D14 (late) samples were plotted for animals with no tumours (Nil), CT26 tumours and 4T1 tumours, displaying all values, as well as box plots with min to max whiskers and means as '+'symbols. These were divided into tumour-specific features and common features between the tumour subtypes. Number of samples was as described in Fig 1B. *p*-values to investigate significance between the cohort means as assessed using 2-way ANOVA on Log (y+0.0001) transformed data using Tukey's multiple comparisons with *, $p \leq 0.05$. **, $p \leq 0.01$. ***, $p \leq 0.001$. and ****, $p \leq 0.0001$.

In contrast, most features associated with 4T1 tumour growth were significantly different from normal levels in Nil animals, with G-CSF level and neutrophil count being >10-fold higher, PD-L1[+] myeloid cell count being ~2-fold higher, and both CXCL13 and IL-6 levels being ~<2-fold higher than normal (Fig 5). B cell number, an important early feature for 4T1 growth modelling, was the only key feature not significantly different from normal levels, although the cells at D14 had a trend of being lower than normal in these mice.

Based on the modelling there were only 2 main features in common contributing to both tumour models' growth, D14 blood neutrophil count and D14 PD-L1[+] myeloid cell count (Fig 5). The unique features associated with each tumour were mostly plasma factors. Overall, the

blood immune phenotype of 4T1-bearing mice was definitively abnormal with a few obvious aberrant immune parameters, while CT26-bearing mice had less drastic changes, making inference of key immune factors more difficult without further study.

## Discussion

In this study we aimed to investigate the utility of a high-throughput multiparameter flow cytometry method, coupled with a machine learning (ML)-based statistical analysis, to screen blood for immune features capable of predicting cancer presence and growth, and also make inferences about underlying cancer-immune biology. Using two syngeneic solid tumour models, a 4T1 breast cancer model and a CT26 colorectal cancer model, our workflow revealed that myeloid factors in the blood, such as neutrophils, monocytes and the levels of the myeloid cell-propagator G-CSF, feature prominently as key determinants of tumour classification (Fig 2). Myeloid cells, specifically neutrophils and PD-L1-expressing myeloid cells, were also common associates of tumour size in both models (Fig 5). Tumour-specific blood immune features were also identified, with elevated levels of G-CSF, IL-6 and CXCL13, and B cell counts associating with prediction of 4T1 growth, while blood CCL17, CXCL10, CXCL1, total myeloid cells, CCL2, Ly6C$^{intermediate}$ monocytes, and IL-10 levels were involved with predicting CT26 tumour growth. Many of these factors have been implicated in cancer progression showing the potential utility of our approach.

With a growing appreciation of immune responses as a hallmark of cancer development, immune phenotyping is becoming an increasingly interesting area of research in cancer management [2]. ML is recognised as an important approach to optimising future cancer diagnosis, prognosis and treatment personalisation, and is ideally suited for interpretating the abundant and complex immune parameters involved [14]. ML approaches can also be used to help make inferences about the underlying biological mechanisms that are modelled for, with the development of model explanatory algorithms [8]. In this study, we have chosen to use the Random Forest model [13] as our learner, since it is flexible (in that it can be used in both classification and regression questions), has in-built feature ranking (to help with feature selection), has fewer overfitting issues than some other models, is relatively interpretable, and performs well in real-life clinical applications compared to other shallow models and more extensive deep learning modalities [15, 16]. The applied Random Forest modelling presented here identified several key blood immune features that, in combination, predicted tumour class (with misclassification of only 4 animals of 130) and size (with moderate to strong linear correlation of predicted to actual current and future tumour sizes). In addition, we used the combination of Random Forest feature ranking [13, 17], SHAP explanatory values [8] and Spearman's-based bivariate correlations [18] to help make inferences about underlying features important for outcomes. Intriguingly, while these factors ranked highly in predictive modelling, and several had significant correlations either directly or indirectly with tumour growth, many did not differ significantly from levels measured in non-tumour bearing animals. This raises the question of potential additive or even synergistic roles for these factors in tumour development; the alternative possibility of chance association, however, cannot be discounted. This latter hypothesis can only be probed with further experimental input, such as blocking and/or knockdown/out studies of the identified key features in *in vivo* studies. While this is beyond the scope of the current study, we note that independent reports support a role for these factors in cancer development and these will be discussed below.

One of the most upregulated factors we identified as a potential early driver of 4T1 growth was G-CSF. Previous observations have shown that 4T1 tumour cells are potent producers of G-CSF [19, 20] and that abrogating G-CSF production significantly diminishes tumour growth

in preclinical breast cancer models [19]. We also showed that elevated neutrophils (annotated CD11b$^+$Ly6G$^+$ cells) strongly correlated with advancing tumours (Fig 5). Previous reports show 4T1 tumour cells induce profound granulocytosis *in vivo* [9, 21] and separate reports reveal a critical role for G-CSF in 4T1 growth and metastasis through changes in granulocyte frequencies (referred to in those reports as myeloid-derived suppressor cells, MDSCs, which can have a CD11b$^+$Ly6G$^+$ phenotype) [22]. Clinically, G-CSF can be significantly higher in the plasma of breast cancer patients and plasma levels correlate with more advanced disease [23], as do blood levels of neutrophils [24]. Intriguingly, IL-6, another early signature of 4T1 growth that we identified, cooperates with G-CSF to promote pro-tumour function of neutrophils [25]. IL-6 is often associated with the tumour microenvironment [26] and clinically, circulating IL-6 level is associated with poor prognosis and low survival rate in patients with breast cancer [27], while IL-6 polymorphisms are linked to increased breast cancer risk [28]. Thus, IL-6 and G-CSF may work in concert on neutrophil function to promote breast cancer growth.

We also identified CXCL13 as another early factor correlating with 4T1 tumour growth, and its role in breast cancer has been widely reported [29–31]. However, published studies are conflicting with regards to its role in the 4T1 model, with support for both pro-tumour activity [32] and anti-tumour activity [33]. Indeed, generally, CXCL13 has been shown to drive growth and invasive signals in many tumours, but also correlates with improved survival in other tumours [34], suggesting a context-dependent role for this cytokine in cancer progression. A further intriguing aspect of CXCL13 biology is that it acts as a chemoattractant for B cells [34], which were also identified as an important feature of 4T1 tumour growth in our analysis. The contribution of B cells in antitumour immunity remains controversial [35], with both pro- and anti- tumour effects. In addition, CXCL13 production from bone marrow endothelial cells occurs in response to IL-6 [36], which is also known to be a B cell differentiation and activation factor [34]. Based on these reports, and our data, we can formulate a model for all these factors that potentiates breast cancer growth (Fig 4K). Here, IL-6, promoted by the tumour microenvironment, may interact in concert with G-CSF to drive neutrophil protumour activity and also production of CXCL13. CXCL13 may then act as a protumour factor and, with IL-6, promote B cell responses which also act on tumour growth. Finally, we have identified a PD-L1-expressing myeloid population, a third top feature of our 4T1 ML model (Fig 4I), which correlates with circulating B cell number and thus may also act indirectly to support tumour growth. While circulating PD-L1-expressing myeloid populations are less well documented than the factors described above, it has been reported that, in lung cancer, treatment with PD-1/PD-L1 blockade response correlated with systemic PD-L1$^+$ CD11b$^+$ myeloid cell frequency, suggesting a potential for stratification based on systemic PD-L1$^+$ myeloid cell subsets [37]. Further study of these cells is warranted.

In the CT26 tumour model, we identified early levels of CXCL1, CCL2 and CCL17 as having important roles in predicting tumour growth as well as similar pairwise correlations with factors directly correlating with tumour growth and one another (Fig 3I), suggesting they played similar roles in this context. CXCL1, is known to promote recruitment and activation of neutrophils [38], premetastatic niche formation [39], tumour invasive potential [40] and tumorigenicity in metastatic colorectal cancer patients [41], and therefore, not surprisingly, serves as a biomarker for poor prognosis. Similarly, CCL2 promotes the recruitment of immunosuppressive tumour-associated macrophages [42], promotes CT26 tumour growth [43] and associates with poor outcomes in metastatic human colorectal cancer [42]. In contrast, CCL17 has been reported to play a complex and somewhat contradictory role in cancer development and progression [44]. CCL17 can promote anti-CT26 tumour immune response [45], and high serum levels are associated with improved survival rates in advanced melanoma patients

[46]. On the other hand, tumour-associated neutrophils can produce CCL17, recruiting CD4 T regulatory cells that promote immune evasion and cancer development in non-small cell lung cancer [47, 48]. It is possible that the location, timing and context of CCL17 expression determines its impact on cancer establishment and progression. Indeed, this may also be the case with CXCL1 and CCL2, since all these three factors associated positively with early correlates of CT26 growth, such as blood myeloid cells and plasma CXCL10 levels, but then also associated negatively with late factors correlating with CT26 growth, such as monocytes with a Ly6G$^-$Ly6C$^{intermediate}$ phenotype and neutrophils (Fig 3).

CXCL10 was identified as an early weak correlate of CT26 growth in our analysis. Clinically, CXCL10 been associated with pro- and anti- tumour responses in colorectal cancer patients [49, 50]. A recent study across 3,763 colorectal cancer patients suggested lower CXCL10 expression was significantly associated with disease spread, recurrence and overall survival, and this association was dependent on other factors such as age and population-based genetic differences [50]. This suggests that CXCL10 expression may have potential as a predictive biomarker in colorectal cancer management, once these variables are taken into account. Similarly, IL-10, a feature involved in prediction of CT26 growth in our analysis, is also associated with colorectal cancer patient prognosis, but in a context dependent manner, being generally lower in patients compared to controls, but higher in patients with poor prognosis [51].

While several myeloid cells were identified as late associates of CT26 growth, the late appearance of monocytes with a Ly6G$^-$Ly6C$^{intermediate}$ phenotype had the strongest association with tumour size (Fig 4). Tumour monocyte subsets are known to have diverse roles in tumour progression [52]. Related to this, CCL2 is a primary recruiter of tumour monocyte subsets [52] and CXCL10 is known to be a monocyte recruitment factor [53]. In our study, early levels of CCL2 and CXCL10 were associated with one another and early CXCL10 levels had a weak correlation with the later appearance of Ly6G$^-$Ly6C$^{intermediate}$ monocytes. Based on these observations and reports by others, a potential model for the role of key blood immune factors identified can be postulated in colorectal cancer development (Fig 3K). Here, early production of CCL2 and CXCL1 may help with shaping the initial myeloid cell compartment in cancer-bearing individuals, which promotes tumour development and production of CCL17 and CXCL10 [54] which in turn modulates recruitment of leukocytes. The early soluble factors may then help shape later tumour associated factors such as IL-10, neutrophils, PD-L1-expressing myeloid cells and Ly6G$^-$Ly6C$^{intermediate}$ monocytes, which play roles in tumour development.

Undoubtedly, our work is limited by the choice of models used to develop the workflow. While murine syngeneic cell line models are among the most widely used tools for studying cancer [55, 56] and have been involved in landmark discoveries [57, 58], there are several limitations to this approach. Cell line-derived models are non-autochthonous, and thus may not have the normal architecture or development that occurs in tumours evolving *de novo*. Indeed, the injection of the cell lines may in itself alter the inflammatory environment in a way that would not be seen in *de novo* tumour growth [59]. The loss of genetic heterogeneity and irreversible changes in gene expression resulting from long-term *in vitro* propagation of tumour cell lines may also mean that we do not observe the same level of intra-individual heterogeneity that is common in human tumours [56, 60]. Furthermore, the use of inbred mouse strains does not reflect the vast inter-individual heterogeneity present in the clinic [56]. While we have attempted to overcome some of these issues by using two distinct and diverse cell lines, there would be obvious benefit to increasing this diversity with additional models, given the resources. Nevertheless, there is clinical evidence to support our findings (as discussed above) and thus our study provides an approach that may work clinically, which is the ultimate goal.

While beyond the scope of this current study, the workflow developed here is now being modified for clinical implementation in cancer patients. This will involve initial high-dimensional screens (using protein arrays and LEGENDScreen[TM] technologies) to identify blood cell and plasma features that may be associated with cancer-specific progression. Key features will then be rationalised in a high-throughput assay/machine learning pipeline analogous to that reported here and used to phenotype blood of cancer patients and closely matched healthy controls to assess capacity to predict patient outcomes over time.

In summary, our work demonstrates the benefit of a high-dimensional data pipeline for the identification of key immune features that interact with tumour development. Our analysis has highlighted the great complexity in the relationship between the immune response and tumour development, where expression of a single molecule may well be insufficient to predict or explain tumour progression. Indeed, it is clear that many immune factors have context-dependent roles in cancer development [34, 44, 61]. With this in mind, we believe a multivariate approach to "biomarker" identification for use in the prognostication and treatment personalisation of cancer is well warranted. Furthermore, we are confident that this work demonstrates the utility of an immune-based workflow in combination with ML to enable identification of context-dependent predictive immune features for the study of tumour outcome. It will be of further interest if such an approach can be utilised to predict treatment outcomes, justifying a role for assessing multivariate immune biomarkers for cancer treatment personalisation.

## Supporting information

**S1 Fig. Gating and population names for leukocyte subsets.** FlowJo software was used to delineate leukocyte populations using manual and boolean gates on concatenated samples with the scheme shown in (**a**) acting as a template for the entire study. FIt-SNE plots from concatenated live CD45[+] samples, generated with default FlowJo setting, were overlaid with each manual gated population to ensure the gating scheme generated similar populations to those generated from the unsupervised approach (**b**), with the manual gate population identified by colour and name (**c**). The process was refined until the two approaches were good approximations of each other, resulting in the manual gates displayed in **a**. Generic and short form names of each population were then assigned based on marker expression and used throughout the manuscript (**d**).
(TIFF)

**S2 Fig. Gating and names for LEGENDplex beads.** FlowJo software was used to delineate LEGENDplex bead populations using manual gates for both the Macrophage/microglial (Mac/Mic) 13-plex LEGENDplex kit (**a**) and the Proinflammatory (Proinflam) 13-plex LEGENDplex Kit (Biolegend) (**b**), which acted as a template for the entire study.
(TIFF)

**S3 Fig. Random Forest learning curve for data set size to model performance evaluation.** Normalised blood immune features (S4 Fig) taken from the 130 animals that had both day 7 and day 14 blood samples (Fig 1B), were used in Random Forest modelling to predict presence of tumour and tumour subtype (targets class being Nil, 4T1 and CT26). Modelling was done on a progressively smaller number of random samples and model performance assessed using cross-validation with a training set of 80% of randomly obtained data and tested on the remaining data and this repeated 100 x. Model performance was assessed by several classification indicators, including area under curve of the receiver operating characteristics (AUC; to assess separability of the classes), classification accuracy (CA; proportion of correct

classification), precision (ratio of correct positive prediction to all predicted positive), recall (ratio of correct positive prediction to actual positive), and F1 score (weighted average of precision and recall) with values being from 0 to 1 (and toward the latter being the best).
(TIFF)

**S4 Fig. Normalised blood immune phenotypes in animal tumour models.** CT26 or 4T1 tumours were grown in female, BALB/c mice and blood immune phenotype determined at D7 and D14, as described in Fig 1. Animals with no tumours (Nil) were used as normal immune phenotype controls. A total of 180 animals were included in the study, and animals divided into the groups indicated in Fig 1B. A 20 μl of blood sample from each animal at each time point was phenotyped for leukocyte populations and plasma analytes (Fig 1). Cell and plasma analytes were reported as fold-changes from the mean of Nil mice or "nil normalised", as described in the *methods* for both the D7 and D14 time points and presented on a log-scale (with numbers of 0-value data points indicated on the axis). Means and SEM are indicated (shown in yellow) and mean equality was tested using ANOVA on Log (y+0.0001) transformed data using Tukey's multiple comparisons correction, with 2-way ANOVA and multiple comparison *p*-values indicated (*, $p \leq 0.05$. **, $p \leq 0.01$. ***, $p \leq 0.001$. ****, $p \leq 0.0001$.). Heatmap summaries of the data highlighting the changes are also shown. Three analytes overlapped in the LEGENDplex kits, namely CCL22, CXCL1 and CCL17, and are labelled with a (1) if from the Mac/Mic panel or (2) if they are from the Proinflam panel.
(TIFF)

**S5 Fig. Tumour classification Random Forest modelling using blood immune phenotype.** Normalised blood immune features (S4 Fig) taken from the 130 animals that had both D7 and D14 blood samples (Fig 1B), were used in Random Forest modelling to predict presence of tumour and tumour subtype (target classes being Nil, 4T1 and CT26). The model was trained on 80% (**a**) and 60% (**b**) of randomly selected data and cross-validated using leave-one-out, and tested using the remaining data. Modelling was done on a progressively smaller number of features, from lowest to highest ranked, based on in-built Random Forest importance for class determination, and the process repeated 3 times. Model performance was assessed by several classification indicators, including area under curve of the receiver operating characteristics (AUC; to assess separability of the classes), classification accuracy (CA; proportion of correct classification), precision (ratio of correct positive prediction to all predicted positive), recall (ratio of correct positive prediction to actual positive), and F1 score (weighted average of precision and recall) with values being from 0 to 1 (and toward the latter being the best). The SHapley Additive exPlanations (SHAP) algorithm feature importance scores for classification using the top-15 features (ranked from highest to lowest) from the SHAP values are shown in (**c**), and show how the feature values impact on classification of each animal cohort, namely healthily control (Nil), CT26-bearing and 4T1-bearing cohorts.
(TIFF)

**S6 Fig. Random Forest modelling to predicting CT26 tumour size and growth using blood immune phenotypes.** Normalised blood immune features (S4 Fig) taken from 48 CT26-bearing animals that had both D7 and D14 blood samples (Fig 1B) were used in Random Forest modelling to predict CT26 tumour size at D14. The model was trained initially on 100%, 80% and 60% of randomised data and cross-validated using leave-one-out (*Train* panels) and tested using the remaining data (*Test* panels). Modelling was done on a progressively smaller number of features, from lowest to highest ranked based on in-built Random Forest importance, and the process repeated 3 times (mean and standard error of mean shown). Model performance was summarised showing the 60%:40%, training:testing split, and equality of test and train

performance score means (using the top assigned features) assessed using ANOVA and shown in the main Fig (Fig 3). The Random Forest rank (RF rank) scores for the top-10 features are shown. Model performance was assessed by several regression indicators, including the error scores, Mean Squared Error (MSE), Mean Absolute Error (MAE) and Root Mean Squared Error (RMSE) (which we hoped to minimise), and the coefficient of determination score R2. D14 tumour size was used as the target using D14 blood samples to assess if blood immune features could predict *current* tumour size (**a**). D14 tumour size was used as the target using D7 blood samples to assess if blood immune features could predict *future* tumour size (**b**). (TIFF)

**S7 Fig. Random Forest modelling to predicting 4T1 tumour size and growth using blood immune phenotypes.** Normalised blood immune features (S4 Fig) taken from 58 4T1-bearing animals that had both D7 and D14 blood samples (Fig 1B), were used in Random Forest modelling to predict 4T1 tumour size at D14. The model was trained initially on 100%, 80% and 60% of randomised data and cross-validated using leave-one-out (*Train* panels) and tested using the remaining data (*Test* panels). Modelling was done on a progressively smaller number of features, from lowest to highest ranked based on in-built Random Forest importance, and the process repeated 3 times (mean and standard error of mean shown). Model performance was summarised showing the 60%:40%, training:testing split and equality of test and train performance score means (using the top assigned features) assessed using ANOVA and shown in the main Fig (Fig 4). The Random Forest rank (RF rank) scores for the top-10 features are shown. Model performance was assessed by several regression indicators, including the error scores, MSE, MAE and RMSE (which we hoped to minimise), and the coefficient of determination score R2. D14 tumour size was used as the target using D14 blood samples to assess if blood immune features could predict *current* tumour size (**a**). D14 tumour size was used as the target using D7 blood samples to assess if blood immune features could predict *future* tumour size (**b**). (TIFF)

**S1 Table. List of antibodies for cell surface labelling.** (DOCX)

**S1 File. All raw data.** (XLSX)

**S2 File. RF pipeline classification.** (OWS)

**S3 File. RF pipeline regression.** (OWS)

**S4 File. MDS pipeline for regression.** (OWS)

## Acknowledgments

We wish to acknowledge Mick Devoy and Dr Harpreet Vohra for their expert help with flow cytometry.

## Author Contributions

**Conceptualization:** David A. Simon Davis, Farhan M. Syed, Ines I. Atmosukarto, Benjamin J. C. Quah.

**Data curation:** Benjamin J. C. Quah.

**Formal analysis:** Benjamin J. C. Quah.

**Funding acquisition:** Benjamin J. C. Quah.

**Investigation:** David A. Simon Davis, Sahngeun Mun, Julianne M. Smith, Ines I. Atmosukarto, Benjamin J. C. Quah.

**Methodology:** Benjamin J. C. Quah.

**Project administration:** David A. Simon Davis, Ines I. Atmosukarto, Benjamin J. C. Quah.

**Supervision:** David A. Simon Davis, Ines I. Atmosukarto, Benjamin J. C. Quah.

**Visualization:** Dillon Hammill.

**Writing – original draft:** David A. Simon Davis, Jessica Garrett, Katharine Gosling, Jason Price, Hany Elsaleh, Farhan M. Syed, Ines I. Atmosukarto, Benjamin J. C. Quah.

**Writing – review & editing:** David A. Simon Davis, Sahngeun Mun, Julianne M. Smith, Dillon Hammill, Jessica Garrett, Katharine Gosling, Hany Elsaleh, Farhan M. Syed, Ines I. Atmosukarto, Benjamin J. C. Quah.

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
