## [Decision Letter · Decision Letter 0]

5 Jan 2022

PONE-D-21-35248Machine learning predicts cancer outcomes from blood immune signaturesPLOS ONE

Dear Dr. Quah,

Thank you for submitting your manuscript to PLOS ONE. After careful consideration, we feel that it has merit but does not fully meet PLOS ONE’s publication criteria as it currently stands. Therefore, we invite you to submit a revised version of the manuscript that addresses the points raised during the review process.The reviewers have asked for adding the limitations of the study as well as some clarifications. Kindly address the reviewer comments and resubmit the manuscript by Feb 19 2022 11:59PM.

Please include the following items when submitting your revised manuscript:A rebuttal letter that responds to each point raised by the academic editor and reviewer(s). You should upload this letter as a separate file labeled 'Response to Reviewers'.A marked-up copy of your manuscript that highlights changes made to the original version. You should upload this as a separate file labeled 'Revised Manuscript with Track Changes'.An unmarked version of your revised paper without tracked changes. You should upload this as a separate file labeled 'Manuscript'.If applicable, we recommend that you deposit your laboratory protocols in protocols.io to enhance the reproducibility of your results. Protocols.io assigns your protocol its own identifier (DOI) so that it can be cited independently in the future. For instructions see: https://journals.plos.org/plosone/s/submission-guidelines#loc-laboratory-protocols. Additionally, PLOS ONE offers an option for publishing peer-reviewed Lab Protocol articles, which describe protocols hosted on protocols.io. Read more information on sharing protocols at https://plos.org/protocols?utm_medium=editorial-email&utm_source=authorletters&utm_campaign=protocols.

We look forward to receiving your revised manuscript.

Kind regards,

Afsheen Raza, PhD

Academic Editor

PLOS ONE

“This work was partially supported by the Radiation Oncology Private Practice Trust Fund, Canberra Health Services.”

Please note that funding information should not appear in the Funding section or other areas of your manuscript. We will only publish funding information present in the Funding Statement section of the online submission form.

“This work was partially supported by the Radiation Oncology Private Practice Trust Fund, Canberra Health Services, Canberra, Australia. The funder provided support in the form of salaries and/or research materials for authors B.J.C.Q, D.A.S.D., S.M., J.S., F.M.S., I.I.A. but did not have any additional role in the study design, data collection and analysis, decision to publish, or preparation of the manuscript.”

“We have read the journal's policy and the authors of this manuscript have the following competing interests: I.I.A., J.P., and K.G. declare that they are employees of the biotechnology company Lipotek Pty Ltd.  The remaining authors have declared that no competing interests exist.”

Reviewers' comments:

Reviewer's Responses to Questions

**Comments to the Author**

1. Is the manuscript technically sound, and do the data support the conclusions?

Reviewer #1: Yes

Reviewer #2: Yes

2. Has the statistical analysis been performed appropriately and rigorously? 

Reviewer #1: Yes

Reviewer #2: Yes

3. Have the authors made all data underlying the findings in their manuscript fully available?

Reviewer #1: Yes

Reviewer #2: Yes

4. Is the manuscript presented in an intelligible fashion and written in standard English?

Reviewer #1: Yes

Reviewer #2: Yes

5. Review Comments to the Author

Reviewer #1: This study is well designed and described. The major limitations include using (1) cell lines, (2) using only two cell lines, and (3) using mouse xenograft models. The authors allude to (1) in the Discussion (last paragraph), but all of these limitations should be further discussed. Additionally, the potential translation of this approach to clinical practice should be discussed in greater detail. What are the steps to get there and how would this workflow be applied to actual patients?

Reviewer #2: This is an excellent study that establish a blood immune signature predicting cancer outcomes by machine learning. But I still have some advice about this study:

1.It’s better to valid the application of this signature in patients.

2.You can add some experiments to certify the result of this study in vivo and in vitro.

6. PLOS authors have the option to publish the peer review history of their article (what does this mean?). If published, this will include your full peer review and any attached files.

Reviewer #1: No

Reviewer #2: No

---

## [Author Response · Author response to Decision Letter 0]

23 Jan 2022

Dear Editor,

Please find below responses to the specific queries by the reviewers. We have modified the manuscript’s discussion to address all of these points with track changes.

Reviewer #1: This study is well designed and described. 

The major limitations include using (1) cell lines, (2) using only two cell lines, and (3) using mouse xenograft models. The authors allude to (1) in the Discussion (last paragraph), but all of these limitations should be further discussed.

Response: We have now extensively broadened the discussion to include a section (from pg 30) on the use of cell lines and murine cancer models, highlighting their limitations and benefits in cancer research. We have also highlighted the benefits of using more cell line models in this context. It should also be noted that xenografts were not used in this study as suggested by the reviewer, only syngeneic models, and so only syngeneic models were addressed in the new discussion section.

Additionally, the potential translation of this approach to clinical practice should be discussed in greater detail. What are the steps to get there and how would this workflow be applied to actual patients?

Response: We have included details of our clinical approach in the above-mentioned new section of the discussion (from pg 30).

Reviewer #2: This is an excellent study that establish a blood immune signature predicting cancer outcomes by machine learning. 

But I still have some advice about this study:

1.It’s better to valid the application of this signature in patients.

Response: We agree very much with this statement and it is indeed what we are currently working towards. The current report was to provide evidence in order to help gain funding and resources to pursue such a human study. Therefore, we feel that to include a human study in this particular report is beyond the scope of the current work. However, we have now included an extensive additional discussion section (as mentioned above from pg 30) highlighting the limitation of our current study and the need for validation in the clinic, and suggesting steps to pursue this.

2.You can add some experiments to certify the result of this study in vivo and in vitro.

Response: While such a general statement is difficult to address specifically, we have modified our discussion section (on pg 26), in which we previously addressed the need for further experimentation to support for our findings, to include a brief detail of the experiments that could be done to support the numerous features we identified that may be involved in cancer development. We note that these experiments would be vast given the number of features involved and are out of the scope of the current study. Instead support our conclusions based on reference to independent peer-reviewed studies that support our findings and which we have summarised in the discussion.

Response: Please note, as requested, we would like to notify the journal that five new references are included in the revised manuscript to help support the changes to the discussion.

Finally, we have taken this opportunity to make some minor amendments to the title, abstract, introduction and discussion, as well as making some minor grammatical changes, which we believe improve the readability of the manuscript without changing the essence of these sections. If this goes beyond what is appropriate, we are happy to exclude these changes and can send this to you instead, please let us know if this is preferred.

Thank you for considering our work for publication in PLOS ONE.

Yours sincerely

Dr Ben Quah

Principal Investigator, I-Cube lab

ACRF Department of Cancer Biology & Therapeutics

The John Curtin School of Medical Research

The Australian National University

---

## [Editor Report · Decision Letter 1]

15 Feb 2022

Machine learning predicts cancer subtypes and progression from blood immune signatures

PONE-D-21-35248R1

Dear Dr. Quah,

We’re pleased to inform you that your manuscript has been judged scientifically suitable for publication and will be formally accepted for publication once it meets all outstanding technical requirements.

Kind regards,

Afsheen Raza, PhD

Academic Editor

PLOS ONE
---

## [Editor Report · Acceptance letter]

17 Feb 2022

PONE-D-21-35248R1 

Machine learning predicts cancer subtypes and progression from blood immune signatures 

Dear Dr. Quah:

I'm pleased to inform you that your manuscript has been deemed suitable for publication in PLOS ONE. Congratulations! Your manuscript is now with our production department. 

Kind regards, 

on behalf of

Dr. Afsheen Raza 

Academic Editor

PLOS ONE